# Multiband charge density wave exposed in a transition metal dichalcogenide

Árpád Pásztor [1✉], Alessandro Scarfato [1], Marcello Spera[1], Felix Flicker[2,3,4], Céline Barreteau [1], Enrico Giannini [1], Jasper van Wezel[5] & Christoph Renner [1✉]

In the presence of multiple bands, well-known electronic instabilities may acquire new complexity. While multiband superconductivity is the subject of extensive studies, the possibility of multiband charge density waves (CDWs) has been largely ignored so far. Here, combining energy dependent scanning tunnelling microscopy (STM) topography with a simple model of the charge modulations and a self-consistent calculation of the CDW gap, we find evidence for a multiband CDW in 2H-NbSe$_2$. This CDW not only involves the opening of a gap on the inner band around the K-point, but also on the outer band. This leads to spatially out-of-phase charge modulations from electrons on these two bands, which we detect through a characteristic energy dependence of the CDW contrast in STM images.

[1] Department of Quantum Matter Physics, Université de Genève, 24 quai Ernest Ansermet, CH-1211 Geneva 4, Switzerland. [2] Department of Physics, Clarendon Laboratory, Rudolph Peierls Centre for Theoretical Physics, University of Oxford, Parks Road, Oxford OX1 3PU, UK. [3] School of Physics and Astronomy, Cardiff University, Cardiff CF24 3AA, UK. [4] School of Mathematics, University of Bristol, Bristol BS8 1TW, UK. [5] Institute for Theoretical Physics Amsterdam and Delta Institute for Theoretical Physics, University of Amsterdam, Science Park 904, 1098XH Amsterdam, The Netherlands. ✉email: arpad.pasztor@unige.ch; christoph.renner@unige.ch

Imposing a new periodicity on a crystal leads to the reorganization of the electronic bands of the parent compound through their back-folding on the new Brillouin zone. New periodicities may be engineered in designer materials, for instance in artificial heterostructures with Moiré minigaps, or emerge due to a structural or electronic phase transition. The charge density wave state is an electronic order where the charge density develops a spatial modulation concomitantly to a periodic distortion of the atomic lattice and the opening of a gap in the quasi-particle spectrum. By reducing the electronic band energy, this gap compensates for the elastic and Coulomb energy costs associated with the formation of the CDW. It also lowers the degeneracy of the electronic states at the crossings of the folded bands. These are the points in the band structure of the parent compound that are connected by the wavevector of the new periodicity. Although a gap should open at all the crossings of the folded bands, previous studies only focused on the primary CDW gap around the Fermi-level, which leads to the highest energy gain of the reconstructed system. The existence of secondary gaps and associated charge modulations (CMs) remains largely unexplored.

In many cases, only a tiny fraction of the electrons are involved in the CDW formation. Therefore, the CDW gap manifests only as a slight reduction of the density of states (DOS)—which can depend on momentum—rather than a full depletion of the DOS. This makes it challenging to measure the CDW gap using spectroscopic probes such as angle-resolved photoemission spectroscopy (ARPES) and scanning tunnelling spectroscopy. This is particularly true for 2H-NbSe$_2$[1–5] (hereinafter simply NbSe$_2$). However, the effect of the redistributed electrons can be readily detected in topographic STM images, even for minute changes brought upon by the opening of the CDW gap as demonstrated in the following.

NbSe$_2$ is an iconic material of correlated electron physics. It hosts a nearly commensurate charge density wave below $T_{CDW} = 33.5$ K and a superconducting (SC) order below $T_{SC} = 7.2$ K[6–10]. NbSe$_2$ is a layered material with a three-fold symmetric crystal structure around the direction perpendicular to the layers (Fig. 1a). Each unit cell is composed of two slabs of Se–Nb–Se trilayers, where the Se lattices are 60° rotated, while the Nb atoms are aligned on top of each other in a trigonal prismatic coordination with the Se atoms.

The Fermi surface (FS) of NbSe$_2$ is mainly determined by the bonding and antibonding combinations of the Nb-4$d$ orbitals[11,12] leading to double-walled barrel-shaped pockets around the K and Γ points of the hexagonal Brillouin-zone[2,3,13,14] (Fig. 1b). The charge-ordered state consists of three CDWs, which form along the three equivalent $\Gamma M$ directions with wavevectors $(1-\delta)\frac{2}{3}|\Gamma M|$, where $\delta \approx 0.02$ and depends on temperature[10]. In real space, this yields a

locally commensurate $3a_0 \times 3a_0$ superstructure delimited by discommensurations[15,16], where $a_0$ is the atomic periodicity.

The $3a_0 \times 3a_0$ reconstruction is readily accessible to topographic STM imaging. Its bias-dependent contrast has been the focus of previous studies, with particular emphasis on the contrast inversion expected in a classic Peierls scenario between images acquired above and below the CDW gap[17,18], and on the role of defects in stabilizing the CDW[5]. Sacks et al.[18] calculate the bias dependence of the CDW phase in a perturbative approach, considering a single band normal state description of NbSe$_2$. They find that the phase-shift of the CDW component of the local DOS can be very different from the 180° expected in a one-dimensional (1D) case (Supplementary Note VII) when changing the imaging bias across the Fermi level ($E_F$). However, their model does not reproduce the full bias dependence of the CDW amplitude and phase that we find.

## Results and discussion

**Bias dependent STM topography, CDW phase and amplitude.** In Fig. 2a–c, we present a selection from numerous topographic STM images of the same region on a cleaved NbSe$_2$ surface at different sample biases ($V_b$) between $-0.5$ V and $0.5$ V. They show a triangular atomic lattice with a superimposed $3a_0 \times 3a_0$ CDW modulation (see also Supplementary Fig. 1a), consistent with previous STM studies of unstrained bulk NbSe$_2$[4,5,16,17,19–22]. A defect-free region with a well resolved CDW outlined in red is magnified in Fig. 2d, f, h for each $V_b$. In order to identify the origin of the bias dependence of the topographic contrast in these images, we separate the atomic lattice and CDW contributions using Fourier filtering (Supplementary Note I). This analysis shows that the bias-dependent STM contrast is due to the changing CDW signal (Fig. 2e, g, i) since the corresponding atomic lattice contrast remains unchanged (see Supplementary Fig. 1).

The observed CDW pattern can be modelled as the sum of three plane waves as described in ref[16]. While each plane wave has its own phase $\varphi_i(\mathbf{r})$, which depends on a selected reference point, the *dephasing parameter* $\Theta(\mathbf{r}) = \varphi_1(\mathbf{r}) + \varphi_2(\mathbf{r}) + \varphi_3(\mathbf{r})$ mod 360° is uniquely defined for each particular CDW pattern, independent of any reference point. $\Theta(\mathbf{r})$ represents the internal CDW structure, quantifying the local relative position of the wavefronts of the three CDWs. In Fig. 2j–l, we show $\Theta(\mathbf{r})$ corresponding to the STM images in Fig. 2a–c, respectively. They were obtained by fitting the CDW contrast following the method described in ref.[16].

Each bias voltage is characterized by a dominant dephasing parameter (Fig. 2j–l), except in the vicinity of defects discussed later. This visual assessment is confirmed by the peaked histograms of $\Theta(\mathbf{r})$ (Supplementary Fig. 2). Fitting a Gaussian to these histograms allows to extract a well-defined dephasing parameter $\Theta_0(V_b)$ for each imaging bias (Supplementary Fig. 3). For a quantitative analysis of the bias dependence of $\Theta_0$, we note that a given local CDW structure is represented by any arbitrary combination of $\varphi_i(\mathbf{r})$ summing up to the same dephasing parameter, in particular the one where all three phases are equal. Moreover, the threefold symmetry of the system implies there is no privileged plane wave among the three used to describe the CDW. These observations allow us to map the problem to a one-dimensional (1D) description with an apparent CDW phase $\varphi_0(V_b) = \Theta_0(V_b)/3$ (Supplementary Note III), and model $\varphi_0(V_b)$ to understand the bias dependent CDW pattern.

Plotting $\varphi_0(V_b)$ in Fig. 3a reveals a striking non-monotonic bias dependence, with an inflexion point around $-0.15$ V and a minimum slightly above the Fermi-level ($E_F = 0$ V). This dependence is robust as long as $\varphi_0(V_b)$ is extracted from

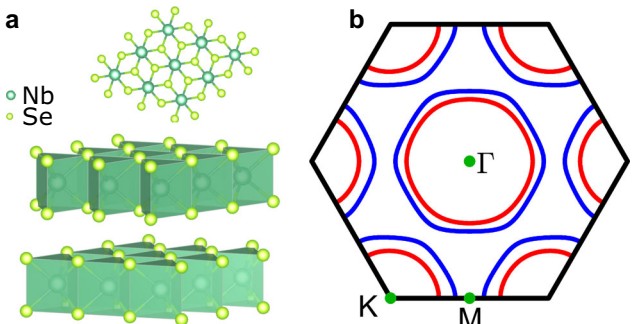

**Fig. 1 Crystal structure and Fermi surface of NbSe$_2$. a** The three-fold symmetric crystal structure from top and side views[33]. **b** The Fermi surface has been calculated using a two-band tight-binding fit to ARPES data[3]. Inner pockets (red) around Γ and K derive from one band, while the outer pockets (blue) derive from the second band; a small pancake-shaped pocket around Γ originating mainly from Se orbitals has been omitted.

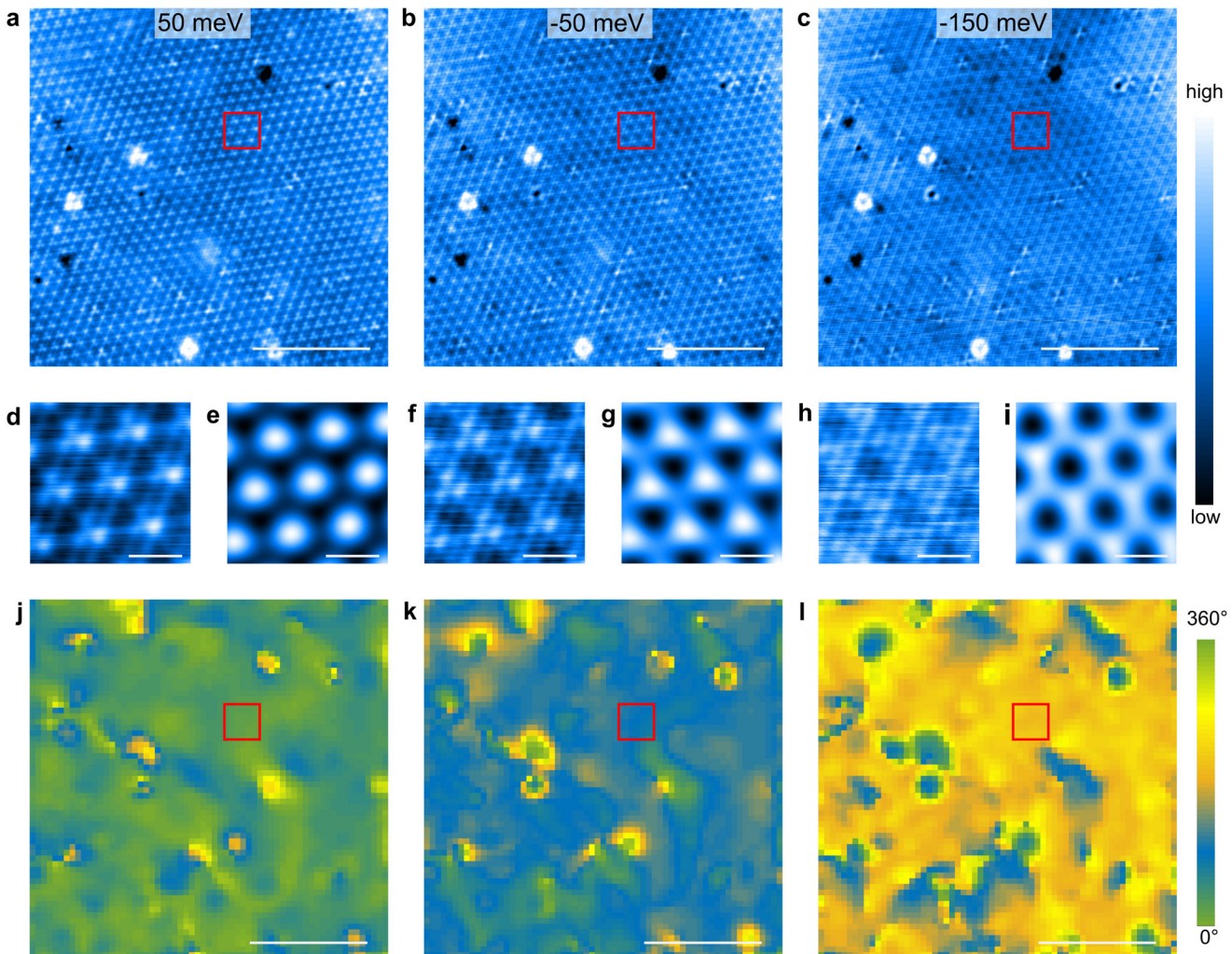

**Fig. 2 Bias-dependent STM imaging of NbSe₂ at 1.2 K.** Constant current topography showing the atomic lattice and CDW at **a** $V_b = 50$ mV, **b** $V_b = -50$ mV and **c** $V_b = -150$ mV (Solely for visualization, the STM image in **c** has been corrected for diagonal running sharp lines (with wavelength much longer than those of the CDW) which arose due to a tiny horizontal tilt of the sample leading to a least significant bit issue in the digital–analogue conversion of the piezo voltage.) with $I_t = 100$ pA. **d**, **f** and **h** are magnified images of the areas marked by the red squares in **a**, **b** and **c**, respectively. The overall imaging contrast is very different in these cases, although the atomic lattice appears identical in all images (Supplementary Fig. 1). **e**, **g** and **i** shows the magnified image of the large Fourier-filtered image of the CDWs at the same location as shown in **d**, **f** and **h**, respectively. It demonstrates that the variation of contrast observed in **d**, **f** and **h** is stemming from a variation in the appearance of the CDW at different biases. These appearances can be quantified by a single parameter: the dephasing parameter which describes the relative position of the three CDWs. **j–l** show the spatial variation of the dephasing parameter determined by fitting the CDW modulations of the STM images shown in **a–c**, respectively. The red squares correspond to the same area that is highlighted in **a–c** and magnified in **d–i**. Scalebars: 10 nm in **a–c** and **j–l**; 1 nm in **d–i**.

topographic STM images away from defects (Supplementary Fig. 4). Close to defects, the dephasing parameter $\Theta_0(V_b)$ is different and tends to depend much less on imaging bias (Supplementary Fig. 5). This is consistent with earlier findings that defects (and impurities) can act as strong pinning centres[23,24] locking the local phase of the CDW or driving the formation of CDW domains[25,26].

The CDW amplitude can be extracted in a similar way to the phase, by fitting the histogram of the local amplitudes $a_i(\mathbf{r})$ of each plane wave measured over the entire field of view with a Gaussian, and extracting the peak value $a_i(V_b)$. The bias dependence and magnitude of $a_i(V_b)$ is nearly the same for all three CDWs (Supplementary Fig. 6b). For the analysis, we consider the average of these three amplitudes at each bias $A_0(V_b) = (a_1(V_b) + a_2(V_b) + a_3(V_b))/3$ plotted in Fig. 3b.

**Modelling and calculations**. To understand the bias dependence of the CDW amplitude and phase in Fig. 3, we simulate

topographic STM traces using a 1D model system (Supplementary Note VIII). In the simplest configuration corresponding to the Peierls reconstruction, we consider the contribution to the tunneling current of a single CM and its associated gap centred on $E_F$ (Fig. 4a–c). In this case, traces at the same polarity are always in phase, whereas traces at opposite polarities always show contrast inversion (or a 180° phase shift in the present harmonic model). The latter, often considered as an identifying hallmark of the CDW state[27], clearly does not reproduce the data in Fig. 3a.

A single CM can only produce two sets of STM traces differing by contrast inversion in the vicinity of the gap. To generate a more complex bias dependence of the phase, we consider the possibility of a second CM whose associated gap opens in another band and away from $E_F$ (Fig. 4d–f). If these two harmonic CMs are in phase (Fig. 4d), the resulting STM traces are either in-phase or 180° out of phase (Fig. 4e), unable to reproduce the data in Fig. 3a. To generate more structures in the bias dependence of the phase, we need to introduce a phase shift between the two CMs

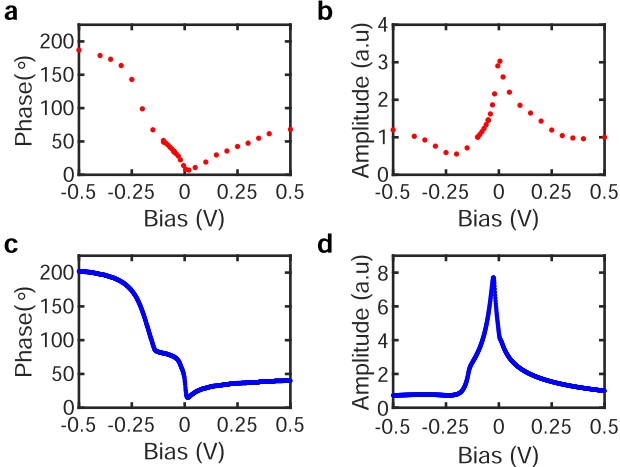

**Fig. 3 Bias dependence of the phase and amplitude in experiments and in modelling. a** and **b** show the bias dependence of the phase and amplitude of one of the unidirectional CDW extracted from the dephasing parameter and amplitude maps. **c** and **d** The phase and amplitude of the best matching simulations in the two-gap model. The data in **b** and **d** are normalized to their $V_b = 0.5$ V value.

(Figs. 4g–i). This leads to a phase that is no longer bi-modal, limited to two values differing by 180° as in Fig. 4c, f. It takes many different values (Fig. 4i), where the precise bias dependence is defined by the magnitude of the two gaps, their position relative to $E_F$ and by the relative phase shift between the two CMs. The simulated STM topographic traces in Figs. 4b, e and h also reveal a pronounced bias-dependent imaging amplitude with distinct line-shapes in the three model cases discussed above (Supplementary Fig. 7).

The broad parameter space of our 1D model makes it challenging to run a self-converging fit to the data. Visually optimizing the size and position of the two gaps in Fig. 4i, we find a range of parameters (Supplementary Note X) simultaneously reproducing the experimental bias dependent CDW phase and amplitude data remarkably well (Fig. 3). As for the relative phase between the two CMs, it is chosen to minimize the Coulomb interaction of the CMs and to conform with the strong commensuration energy that locally locks them to the lattice. Reducing the Coulomb energy is obtained by introducing a phase shift between the two CMs, which can only be ±120° (Fig. 4g) to satisfy the lock-in criterion with the lattice given the $3a_0$ periodicity of the CDW.

In the following, we turn to theoretically modelling multiple CDW gaps on different bands in NbSe$_2$. We deploy self-

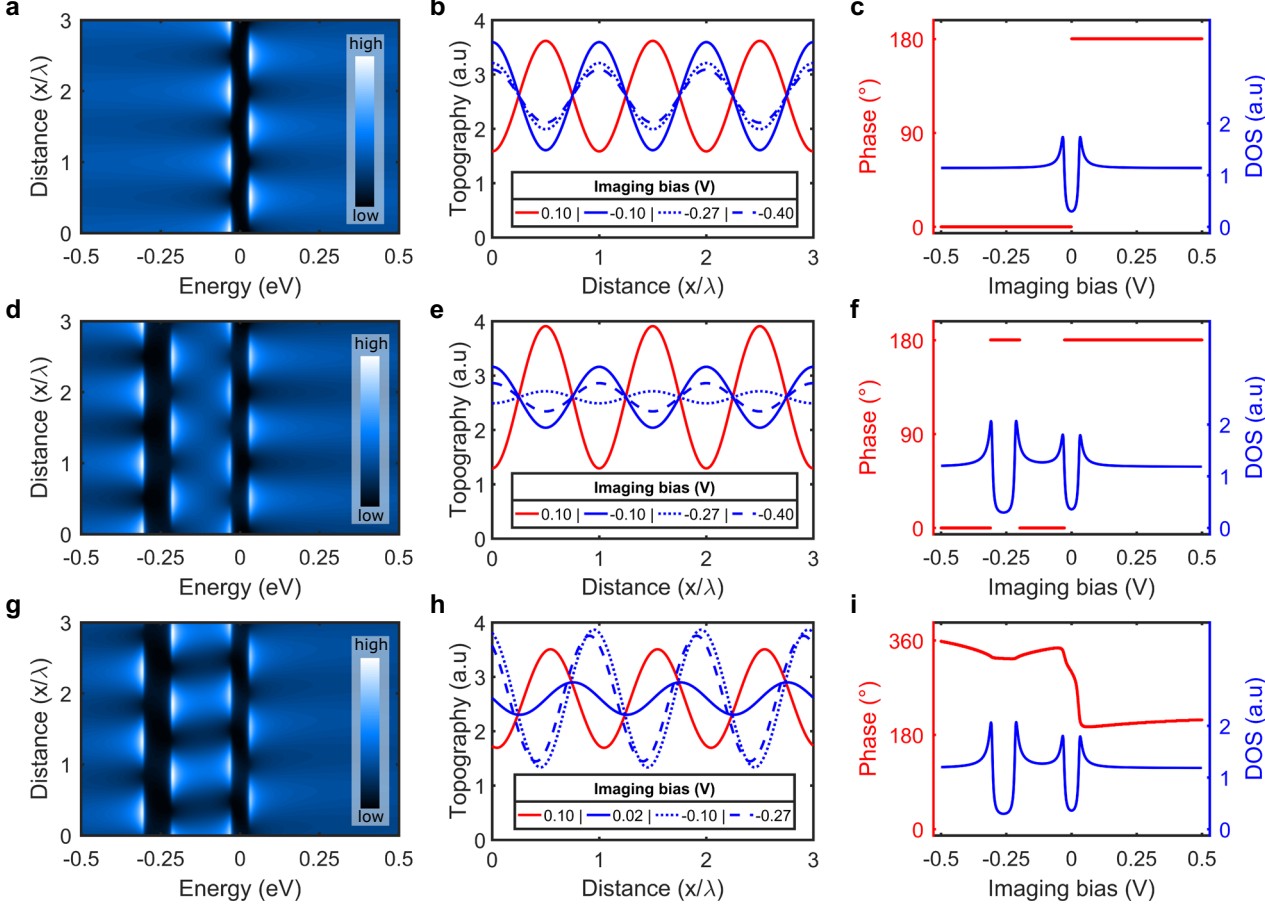

**Fig. 4 One-dimensional model description of the bias-dependent phase of the CDW. a–c** There is a single CM and the corresponding gap is centred at the Fermi level. **d–f** There are two CMs with two gaps that are centred at different energies (one at the Fermi-level and one below). There is no real-space phase difference between the CMs. **g–i** The same as **d–f** except that there is a 120° ($2\pi/3$) phase difference -one atom shift- between the two CMs. In all three cases, the first column shows the spatial and energy-dependent CDW local DOS maps. Second column: corresponding simulated topographic traces at selected biases (the actual bias value is shown in the legend). The red and blue topographic traces correspond to positive and negative sample biases, respectively. To clearly see the evolution of the phase and the amplitude the curves are offset vertically in **b**, **e** and **h** such that they all oscillate around the same value. Third column: bias dependence of the phase (left red axis) and DOS (spatially integrated local DOS from the first column, right blue axis).

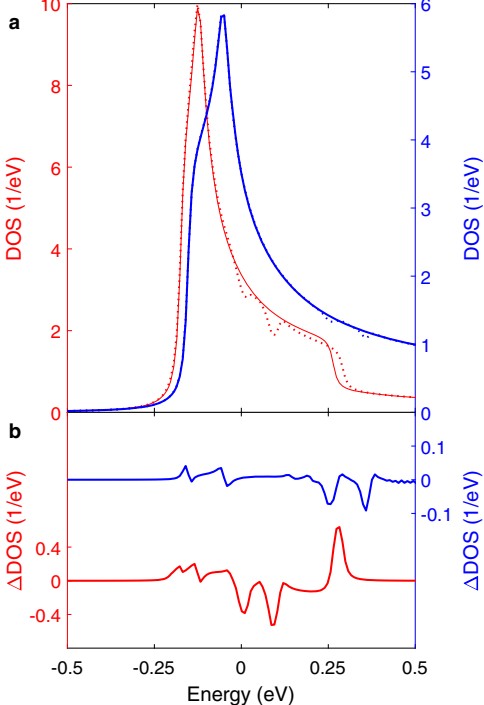

**Fig. 5 Density of state of the two bands in the self-consistent calculations. a** The red (blue) indicates the band making up the inner (outer) pockets; solid/dashed indicates the ungapped/gapped band structure, where gaps are included self-consistently at the mean-field level. **b** Difference between the gapped and ungapped cases in **a**, showing small DOS suppressions at energies away from $E_F$ on both bands in addition to the gap at $E_F$ on the inner band.

consistent calculations to include the CDW gap within the random phase approximation on the two-dimensional two-band tight-binding fit to the NbSe₂ band structure constrained by ARPES[3]. The corresponding FS shown in Fig. 1b consists of inner (red) and outer (blue) bands originating from symmetric and antisymmetric combinations of the Nb $d_{3z^2-r^2}$ orbitals. The model (see "Methods") was previously shown to accurately reproduce the full range of experimental measurements on the charge-ordered state[12,28]. The resulting DOS for the gapped and ungapped cases in each band are shown in Fig. 5a. To emphasize the DOS reduction accompanying the CDW phase transition, we plot the difference between the gapped and ungapped DOS for each band in Fig. 5b.

Our theoretical modelling shows a clear gap on the inner band at $E_F$, consistent with the gap measured by ARPES around the $K$-point[3]. Interestingly, Fig. 5b reveals further DOS reductions, for example near 100 meV on the inner band and −50 meV on the outer band. These features are indicative of CDW gaps opening away from $E_F$ in addition to the (primary) gap at $E_F$, supporting the simple model we propose to understand the bias dependence of the CDW appearance in STM images of NbSe₂. According to Fig. 5b, there could even be more than two gaps. Consequently, we have included up to $n = 8$ gaps to our 1D model. However, the agreement with the data is similar for $n = 2$ and $n = 3$ (Supplementary Note XI), and we see no improvements adding more gaps.

In summary, the remarkable match between the bias dependence of the CDW contrast in STM topography and the simple 1D model proposed here provides compelling evidence that the CDW in NbSe₂ is composed of at least two out-of-phase CMs on the inner and outer bands. While a 180° phase shift between these two CMs would minimize the Coulomb energy, the complex bias dependence of the CDW amplitude and phase observed by STM can only be

reproduced when considering also the commensuration energy. This highlights the importance of the coupling of charge order to the lattice, which manifests in the formation of discommensurations[15,16] and ultimately enables the observation of the multiband CMs uncovered here. The present study further highlights the power of topographic imaging to gain unique insight into detailed features of the CDW too faint to be detected accurately by tunneling spectroscopy. The formation of multiple modulations in response to new periodicities of a primary transition directly observed here, is extremely general and should in principle be present in all charge (and spin) density wave materials, and suggests new directions to explore in the physics of spatially modulated electronic orders.

## Methods

**Crystal growth and STM measurements**. Single crystals of NbSe₂ were grown via iodine-assisted chemical vapour transport and cleaved in-situ at room temperature. STM experiments were done in UHV (base pressure below $2 \cdot 10^{-10}$ mbar) using tips mechanically cut from a PtIr wire and conditioned in-situ on a clean Ag(111) single crystal. The bias voltage was applied to the sample. STM images were recorded in constant current mode with at least 64 pixel/nm resolution. Details of the CDW amplitude and phase fitting procedure can be found in ref [16].

**Self-consistent calculations**. We carried out a 22-orbital Slater–Koster tight-binding fit[29] to the NbSe₂ band structure, constrained by ARPES measurements[2,3] and local density approximation numerical calculations[11]. This provided not only the band structure, but the orbital composition of the bands. We found that the two bands crossing the Fermi level are predominantly composed of the niobium $d_{3z^2-r^2}$ orbitals (>60% across the Brillouin zone). A third, small, pancake-shaped pocket centred on Γ derives primarily from the selenium p-orbitals and is therefore not expected to mix significantly with the other bands, in-keeping with experimental observations that it plays no role in the charge ordering. We then re-fit the two bands of interest using only the two relevant orbitals; the fit was indistinguishable from the phenomenological fit to ARPES data provided in ref. [3].

The Coulomb interaction can be neglected in NbSe₂, as the large DOSs at the Fermi level leads to strong screening. This remains true down to the lowest temperatures (above the SC transition at 7.2 K) since the CDW gap only opens on small regions of the FS. The relevant interaction is the electron–phonon coupling, for which we constructed an analytic expression following Ref. [30]. It has long been suggested that the CDW in NbSe₂ originates not from FS nesting, but from the dependence of the electron–phonon coupling on the momentum transfer in the phonon-mediated electron–electron scattering[11]. Our calculation includes the dependence of the coupling on the ingoing and outgoing electron momenta, as well as the orbital composition of the electronic bands scattered between. Only by taking all of these factors into account were we able to find a consistent explanation of the full range of experimental observations, including ARPES[2,3], scanning tunneling spectroscopy/microscopy[4], and inelastic X-ray scattering[31]. Our model has only one free parameter, the overall magnitude of the electron–phonon coupling, which we fixed using $T_{CDW} = 33.5$ K.

We modelled the effect of the CDW on the electronic band structure using the random phase approximation. We employed the Nambu–Gor'kov method to work within the gapped phase; this method consists of promoting the electronic Green's function to a $9 \times 9$ matrix, representing the tripling of the real-space unit cell in both lattice directions induced by the CDW formation. The CDW gap appears in off-diagonal elements, and diagonalisation then results in a gapped electronic band structure. We solved for the CDW gap self-consistently at high-symmetry points in the Brillouin zone, and used the results to constrain a tight-binding fit for the gap structure. We assumed the gap to be independent of energy. The CDW gap serves as an order parameter, and so our model naturally accounts for long-range CDW correlations. Further details of the method are given in refs. [12,28].

We found that a CDW gap opens at the Fermi level on the outer pocket centred on the K-point, along the MK line, in agreement with ARPES[2,3]. However, since the order parameter is non-zero at all points in the Brillouin zone, and at all energies, gaps also open wherever bands crossings are introduced. This is the origin of the multiband CDW, evidenced by the suppression of DOS at energies below $E_F$ seen in Fig. 5. We calculated the DOS at different energies, with and without the CDW gap, by summing over the Brillouin zone the spectral function found from the electronic Green's function.

## Data availability

The data that support the findings of this study has been deposited in the Yareta repository (https://doi.org/10.26037/yareta:en553fvpkrdorgqtssxegl4tyi)[32].

## Code availability

Computer codes are available upon reasonable request and preferably within a collaboration.

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

## Acknowledgements

We acknowledge A. F. Morpurgo and I. Maggio-Aprile for inspiring discussions. We thank A. Guipet and G. Manfrini for their technical support in the STM laboratories. This project was supported by Div.2 (Grant No. 182652) and Sinergia (Grant No. 147607) of the Swiss National Science Foundation. F. F. acknowledges support from the Astor Junior Research Fellowship of New College, Oxford.

## Author contributions

C.R. designed the experiment. M.S. and A.S. took care of the STM experiments. Á.P. performed the data analysis. F.F. and J.v.W. performed the self-consistent calculations. Á.P. did the one-dimensional model simulations. C.B. and E.G. synthesized the bulk crystals. Á.P., A.S., F.F., J.v.W. and C.R. wrote the paper. All authors contributed to the scientific discussions and manuscript revisions.

## Competing interests

The authors declare no competing interests.
