## [Peer Review File · Nature Communications]

Reviewer comments & decisions:

Reviewer comments, first version:

Reviewer #1 (Remarks to the Author: Overall significance):

The authors utilize scanning tunneling microscopy, in combination with theoretical approaches to understand the charge density waves in the prototypical system of NbSe₂. In contrast to the established opinions of this long studied system, they claim that the origin of the ordering stems from the formation of two CDWs originating from two bands. They substantiate this claim, by spatially dependent imaging of the CDW and relating this to a two-gap model. While the claim is certainly impactful, should it be substantiated, I do not believe that the evidence in the manuscript proves the main claims of this paper. Likewise, as neither the material, nor the methods used to study this paper do not present a new state of the art, I cannot recommend this for publication in Nature Physics.

From the experimental standpoint, my main issue with the paper has to do with the quality of the material. The images shown in Fig. 2 show a large defect density in the crystal, with the presence of both surface and many subsurface defects. It has been shown that defects in NbSe₂ can lead to a change in the observed CDW (e.g. see PNAS 2013 110 (5) 1623-1627). In order to make the claims the authors are making, it is vital to study a cleaner and more defect free crystal, in order to exclude that modifications of the CDW stem from modulations of the electronic structure stemming from such defects, e.g. electronic disorder. Considering this materials has been studied for decades, and in better quality, understanding this point is essential if the authors which to claim their experiments refute the previous understandings.

Moreover, theoretically, I found myself questioning if the theoretical treatment is accurate enough to treat the problem. In the case, that one is interested in looking at multi-band CDW formation, then I would expect it is necessary to accurately describe long-range correlations in this system. The latter is extremely challenging theoretically, requiring advanced methods (e.g. dual boson, etc.), which may lead to small but important corrections to the electronic structure (e.g. Fig. 5). The authors make no appeal as to how correlations are accurately described, and why more sophisticated treatments can be neglected (contrary to literature). I found next to no description of the theoretical methods in the paper or supplement to help substantiate this. While I do not insist that such computations need to be performed, as this is unrealistic, the authors should properly address their approach and why or why not these other approaches are not relevant, or what they expect if such computations were to be performed.

If the authors are able to study a clean material and substantiate the above points concerning the theory, this may be reconsidered for Nature Physics. Otherwise, I believe this paper presents more doubt that makes a breakthrough, and I would otherwise only consider this for Communication physics,

if the claims are relaxed and the points properly addressed.

Reviewer #1 (Remarks to the Author: Strength of the claims):

see above

Reviewer #1 (Remarks to the Author: Reproducibility):

see above

Reviewer #2 (Remarks to the Author: Overall significance):

The manuscript by Renner et al. reports an experiment and theory collaboration regarding on multiple charge density wave (CDW) on surface of 2H-NbSe₂. The work is largely based on their previous works on the same material. In this work, they developed a mapping method from two dimensional charge orders (three-fold CDWs) to one-dimensional one and then extract phase and amplitude of the CDW. From phenomenological fitting, they conclude that observed scanning tunneling spectroscopic charge modulations can be explained by including at least two different charge orders, one at the Fermi energy and the other at the different energy.

As the authors stressed in the manuscript, I also agree that their study on the multiple CDWs with different energetic positions is first among many. Usually, as the authors mentioned, the multiple CDW orders have been discussed in the context of symmetry or multiple Q (nesting momentum) order. Hence, I would like to value the new claim on the multiband CDW very highly. However, the work, specially, analysis is quite phenomenological and the persuasive theoretical confidence compared with multiband superconductivity is lacking. So, in the process of the Guided Open Access program, I recommend the work be suitable for publication in Nature Communication or others, not in Nature Physics.

This system is quite simple and electron-phonon interaction or other physical quantities related with CDWs can be computed in the level of atomistic calculations. If the CDW orders are indeed multiband origins, the author can compute and can show the gap openings at some specific electronic energy bands as well as phononic dispersion from first-principles. Since the energy dependent self-consistent

formulation for electron-phonon coupling similar to Eliashberg formalism for the multiband superconductivity can compute in principle, the conclusive theoretical evidence for multiband CDW can be obtain with a similar approach.

Actually, I found no evidence or theoretical reasons why another CDW order occurs at -0.25 eV since the authors provide only phenomenological fitting data. So, later, it would be very persuasive and decisive work if the authors can show these data. For the moment, the concept of multiband CDWs is new and interesting but solid support for the claim is lacking.

Reviewer #2 (Remarks to the Author: Reproducibility):

N/A

Reviewer #3 (Remarks to the Author: Overall significance):

The manuscript by Pasztor et al. reports on real-space evidence for the multiband nature of 2D charge density wave (CDW) order in an archetypal dichalcogenide material 2H-NbSe₂. By systematic model analysis of atomic-scale topography data taken with cryogenic scanning tunneling microscopy (STM) at various tip-sample bias voltages, particularly how the CDW modulations in the form of topographic STM image contrast depend on bias voltage, this work presents direct and detailed agreement of the STM topography data with a well-grounded CDW model based on three CDWs that are commensurate with the atomic lattice and on two de-phased charge modulations (CM) arising from two principal energy bands. The analytical methodology used is original, in that it extends well-established CDW formalism to multiple energy bands. The key results reported are of both breakthrough and general significance, in demonstrating a novel technique to extract real-space phase information about spatially-modulated electronic orders. Furthermore, the revelation of CDW energy gaps well away from the Fermi level represents crucial evidence for strong electron correlations in the formation of CDW order in 2H-NbSe₂.

Reviewer #3 (Remarks to the Author: Impact):

In my opinion, this manuscript warrants appearance in Nature Physics, because its results shed new light on a widely-studied though still-enigmatic CDW material, and also because the novel methodology it demonstrates can be generally applied to a variety of complex 2D materials as well as nanoengineered heterostructures (epitaxial, van der Waals or Moire pattern).

Reviewer #3 (Remarks to the Author: Strength of the claims):

This work is convincing, in terms of its phenomenological clarity that connects the STM topography data to the multiband CDW model. Of particular merit is the transparent choice of model parameters, which are either constrained by the specific lattice structure of 2H-NbSe₂ (via the tight-binding band structure) or based on general physical considerations (especially that the phase shift between charge modulations of the two bands is determined by the minimization of Coulomb-repulsion as well as lattice-commensuration energies).

Reviewer #3 (Remarks to the Author: Reproducibility):

To further enhance the manuscript's general significance and clarity, the authors may consider elaborating on the following:

- 1) Experimentally, was any temperature dependence observed in the STM image contrast dependence on bias, both below and even above the CDW ordering temperature of 2H-NbSe₂? Theoretically, would one expect temperature to affect the relative strengths between the Coulomb-repulsion and lattice-commensuration energies?
- 2) What justifies neglecting the "pancake" Fermi surface (FS) sheet (centered near the Gamma point) in the multiband CDW model analysis? Was it simply because of its very small phase space relative to the other FS sheets? Or perhaps the 3D geometry of the pancake FS sheet, in contrast to the 2D tubular FS sheets, renders it less conducive to the in-plane CDW order?
- 3) What does the appearance of energy gaps away from the Fermi level imply about the role that strong electron correlations play in the formation of CDWs in 2H-NbSe₂, since band structure calculations by Johannes, Mazin and Howells (Ref. 11) had shown that conventional mechanism based predominantly on Fermi surface nesting does not adequately explain the observed CDW order.
- 4) For clarity in the caption of Figure 4, specify the color coding of the middle column of panels (b, e, h) in order to disambiguate it from that of the right column of panels (c, f, i).
- 5) In Line 243, perhaps replace the word "backing" by "supporting".

Author rebuttal, first version:

“Multiband charge density wave exposed in a transition metal dichalcogenide” – Response to Reviewers

Reviewer#1 :

Overview

The authors utilize scanning tunneling microscopy, in combination with theoretical approaches to understand the charge density waves in the prototypical system of NbSe₂. In contrast to the established opinions of this long studied system, they claim that the origin of the ordering stems from the formation of two CDWs originating from two bands. They substantiate this claim, by spatially dependent imaging of the CDW and relating this to a two-gap model. While the claim is certainly impactful, should it be substantiated, I do not believe that the evidence in the manuscript proves the main claims of this paper. Likewise, as neither the material, nor the methods used to study this paper do not present a new state of the art, I cannot recommend this for publication in Nature Physics.

We thank Reviewer#1 for their very positive assessment that our study offers new and “impactful” insight into the origin of CDW ordering in NbSe₂. We would like to point out that the phrasing “In contrast to the established opinions of this long studied system” suggests a misunderstanding: we are not contradicting all previous studies, and we are not claiming or commenting on whether two or more CDWs must form. Our study is providing evidence that they do form, and we are pointing out that this is actually expected generically but has not been observed so clearly before.

Comment 1

From the experimental standpoint, my main issue with the paper has to do with the quality of the material. The images shown in Fig. 2 show a large defect density in the crystal, with the presence of both surface and many subsurface defects. It has been shown that defects in NbSe₂ can lead to a change in the observed CDW (e.g. see PNAS 2013 110 (5) 1623-1627). In order to make the claims the authors are making, it is vital to study a cleaner and more defect free crystal, in order to exclude that modifications of the CDW stem from modulations of the electronic structure stemming from such defects, e.g. electronic disorder. Considering this materials has been studied for decades, and in better quality, understanding this point is essential if the authors which to claim their experiments refute the previous understandings.

Reviewer#1's main criticism is the supposedly poor quality of the NbSe₂ crystals we studied, with too many defects that would affect the CDW we observe by STM. This assessment is unfounded, as a direct comparison with published data shows. In Table 1, we compile the defect density determined from the collection of published STM images of NbSe₂ shown in Response Figure 1. The defect density in most of them is very similar to what we find in our own STM images. Note that we can find much lower defect densities—similar to that in Ref 5 in Table 1—in selected smaller regions, of the same size as that used in Ref 5. An example is the 18.5 × 18.5 nm² square outlined in orange in Response Figure 1.

No. Ref	Reference	Number of defects	Area (nm ²)	Defect density (1/nm ²)
	Present study	53	33.6 x 33.6	0.047
1	PNAS 2018;115(27):6986-90.	82	30.6 x 30.6	0.088
2	PRL 2019;122(1):016403.	75	50 x 50	0.030
3	PRL 2015;114(2):026802.	63	32 x 32	0.062
4	PNAS 2013;110(5):1623-7.	22	22 x 22	0.045
5	Nat. Comm. 2015 17;6(1):1-7.	6	18.5 x 18.5	0.018
6	Nano Lett. 2019;19(5):3027-32.	8	14 x 14	0.041
7	Nat. Phys. 2015;12:92.	8	13.3 x 13.2	0.046
Average defect density (1/nm ²):				0.047

Table 1: Compilation of defect densities observed in topographic STM images of NbSe₂ throughout the literature. The corresponding STM images are shown in Response Figure 1.

Response Figure 1: Published topographic STM images of NbSe₂ taken from the references listed in Table 1. They are displayed to scale, with an orange rectangle outlining a $18.5 \times 18.5 \text{ nm}^2$ region with lower defect density in our data.

Reviewer#1 is further mistaken when citing PNAS 2013 110 (5) 1623-1627 to support their comment about the impact of defects on the CDW. The modified CDW order discussed in that reference is the consequence of strain, not defects. PNAS 2013 110 (5) 1623-1627 does not discuss “modifications of the CDW” due to “modulations of the electronic structure stemming from such defects, e.g. electronic disorder”, but due to strain, in contradiction to Reviewer#1’s statement.

Comment 1 implies that we do not discuss the role of defects. In fact, we discuss this at length in sections IV and V of the supplementary material. Although defects may be worth a dedicated investigation on their own, we already demonstrate, in section V, how defects pin the CDW phase in their vicinity. For example, while different defects affect the bias dependence of the phase in distinct ways, the observed effects are strikingly similar in the vicinity of some apparently identical defects. Nevertheless, for the purpose of the present study, we explicitly exclude these regions from our analysis. In section IV, we furthermore demonstrate that the bias dependent phase leading to the proposal of a multiband CDW is systematic and perfectly reproducible across the sample surface away from defects, thereby demonstrating the ro-

bustness and validity of our study.

Comment 2

Moreover, theoretically, I found myself questioning if the theoretical treatment is accurate enough to treat the problem. In the case, that one is interested in looking at multi-band CDW formation, then I would expect it is necessary to accurately describe long-range correlations in this system. The latter is extremely challenging theoretically, requiring advanced methods (e.g. dual boson, etc.), which may lead to small but important corrections to the electronic structure (e.g. Fig. 5). The authors make no appeal as to how correlations are accurately described, and why more sophisticated treatments can be neglected (contrary to literature). I found next to no description of the theoretical methods in the paper or supplement to help substantiate this. While I do not insist that such computations need to be performed, as this is unrealistic, the authors should properly address their approach and why or why not these other approaches are not relevant, or what they expect if such computations were to be performed.

A complete theoretical description of the CDW phase transition is yet to be developed for most real 2D and 3D materials. As explicitly acknowledged by Reviewer#1, the computation of multiband CDW formation in a quantitatively realistic model is well beyond the scope of the present study.

Moreover, in the present study, the observed energy dependence of the CDW phases directly points towards the presence of multiband CDW order. The theoretical analysis is not meant to give a quantitative model for this effect. Rather, it is meant to show the qualitative mechanism by which band folding associated with the CDW reconstruction naturally leads to the formation of (one or more) secondary gaps. The fact that this effect can be seen in a purposely simple theoretical model underlines its generic applicability. The simplicity of the model is thus a particular strength of the current analysis, as it directly suggests the same mechanism will be at play in many other CDW materials with multiple bands crossing the Fermi level. It also shows that the qualitative conclusions are independent of “small corrections”, or any particular choice of microscopic theory. Instead, the use of a simple and broadly applicable analysis establishes the generality of our results, and shows its appropriateness for the broad appeal of Nature Physics.

The expectation of the referee that long-range correlations are required to treat the problem may well be true. In fact, since we use a mean-field treatment, we include infinite-range correlations of the order parameter by definition. More advanced methods may be used to include long-range inter-

actions (rather than correlations), but as Reviewer#1 already indicates, these are only expected to lead to “small corrections” in the description of NbSe₂, without altering the qualitative picture that the theoretical model is intended to provide.

For completeness, we include a more detailed description (see below) of our theoretical model in the methods section of the revised manuscript, including a discussion of its previous uses in the literature, and the match it provides with various types of experimental observations. Here, we also emphasize our choice for a simple, yet broadly applicable method, with correspondingly generic appeal.

We thank the referee for providing the opportunity for us to expand on the description of our theoretical model. In the new methods section we make it clear that our model incorporates long-range correlations by construction.

We additionally considered a range of extensions to the model we describe in the paper. Using the mode-mode coupling approximation [C. M. Varma and A. L. Simons, *Phys. Rev. Lett.* 51, 138 (1983); J. E. Inglesfield, *J. Phys. C* 13, 17 (1980); H. Yoshiyama, Y. Takaoka, N. Suzuki, and K. Motizuki, *J. Phys. C* 19, 5591 (1986)] we included electronic self-energies self-consistently. This method includes long-range interactions in a more substantial manner than the random phase approximation in the main text. While mode-mode coupling is important to gain a more accurate estimate of the temperature dependence, we found that this treatment results in little effect on the predicted CDW gap, or the gapped electronic band structure [F. Flicker and J. van Wezel, *Phys Rev B* 94, 235135 (2016)]. We also considered the effects of strain and the possibility of alternative CDW geometries using a 4th order Landau free energy expansion, but again found that neither were relevant to the present study [F. Flicker and J. van Wezel, *Phys Rev B* 92, 201103(R) (2015)]. These more detailed treatments do not affect the qualitative predictions of our model, and so do not affect the conclusion that our phenomenological 1D model accurately reproduces the relevant physics. We did not feel these more complex theoretical approaches were sufficiently relevant for inclusion in the manuscript.

In our revised manuscript we provide more details of our theoretical model by replacing the **Self-consistent calculations** paragraph of the method section with the following revised description.

We carried out a 22-orbital Slater-Koster tight-binding fit [31] to the NbSe₂ band structure, constrained by ARPES measurements [2,3] and local density approximation numerical calculations [11]. This provided not only the band structure, but the orbital composition of the bands. We found that the two bands

crossing the Fermi level are predominantly composed of the niobium $d_{3z^2-r^2}$ orbitals ($> 60\%$ across the Brillouin zone). A third, small, pancake-shaped pocket centred on Γ derives primarily from the selenium p -orbitals and is therefore not expected to mix significantly with the other bands, in-keeping with experimental observations that it plays no role in the charge ordering. We then re-fit the two bands of interest using only the two relevant orbitals; the fit was indistinguishable from the phenomenological fit to ARPES data provided in Ref. [3].

The coulomb interaction can be neglected in NbSe₂, as the large density of states at the Fermi level leads to strong screening. This remains true down to the lowest temperatures (above the superconducting transition at 7.2 K), since the CDW gap only opens on small regions of the Fermi surface. The relevant interaction is the electron-phonon coupling, for which we constructed an analytic expression following Ref. [32]. It has long been suggested that the CDW in NbSe₂ originates not from Fermi surface nesting, but from the dependence of the electron-phonon coupling on the momentum transfer in the phonon-mediated electron-electron scattering [11]. Our calculation includes the dependence of the coupling on the ingoing and outgoing electron momenta, as well as the orbital composition of the electronic bands scattered between. Only by taking all of these factors into account were we able to find a consistent explanation of the full range of experimental observations, including ARPES [2,3], scanning tunneling spectroscopy/microscopy [4], and inelastic x-ray scattering [33]. Our model has only one free parameter, the overall magnitude of the electron-phonon coupling, which we fixed using $T_{\text{CDW}} = 33.5$ K.

We modelled the effect of the CDW on the electronic band structure using the random phase approximation. We employed the Nambu-Gor'kov method to work within the gapped phase; this method consists of promoting the electronic Green's function to a 9×9 matrix, representing the tripling of the real-space unit cell in both lattice directions induced by the CDW formation. The CDW gap appears in off-diagonal elements, and diagonalisation then results in a gapped electronic band structure. We solved for the CDW gap self-consistently at high-symmetry points in the Brillouin zone, and used the results to constrain a tight-binding fit for the gap structure. We assumed the gap to be independent of energy. The CDW gap serves as an order parameter, and so

our model naturally accounts for long-range CDW correlations. Further details of the method are given in Refs. [12,30].

We found that a CDW gap opens at the Fermi level on the outer pocket centred on the K-point, along the MK line, in agreement with ARPES [2,3]. However, since the order parameter is non-zero at all points in the Brillouin zone, and at all energies, gaps also open wherever band crossings are introduced. This is the origin of the multiband CDW, evidenced by the suppression of DOS at energies below E_F seen in Fig. 5. We calculated the DOS at different energies, with and without the CDW gap, by summing over the Brillouin zone the spectral function found from the electronic Green's function.

Comment 3

If the authors are able to study a clean material and substantiate the above points concerning the theory, this may be reconsidered for Nature Physics. Otherwise, I believe this paper presents more doubt that makes a breakthrough, and I would otherwise only consider this for Communication physics, if the claims are relaxed and the points properly addressed.

Reviewer#1 considers our manuscript suitable for Nature Physics, provided we study a clean material and substantiate our theoretical approach. We are confident our responses above fulfill their expectations: we demonstrate that our crystals are as good as they can be, we carefully address the issue of defects in the supplementary material, and we provide additional insight into our theoretical approach in the revised manuscript.

Reviewer#2 (Remarks to the Author):

Overview

The manuscript by Renner et al. reports an experiment and theory collaboration regarding on multiple charge density wave (CDW) on surface of 2H-NbSe₂. The work is largely based on their previous works on the same material. In this work, they developed a mapping method from two dimensional charge orders (three-fold CDWs) to one-dimensional one and then extract phase and amplitude of the CDW. From phenomenological fitting, they conclude that observed scanning tunneling spectroscopic charge modulations

can be explained by including at least two different charge orders, one at the Fermi energy and the other at the different energy.

As the authors stressed in the manuscript, I also agree that their study on the multiple CDWs with different energetic positions is first among many. Usually, as the authors mentioned, the multiple CDW orders have been discussed in the context of symmetry or multiple Q (nesting momentum) order. Hence, I would like to value the new claim on the multiband CDW very highly. However, the work, specially, analysis is quite phenomenological and the persuasive theoretical confidence compared with multiband superconductivity is lacking. So, in the process of the Guided Open Access program, I recommend the work be suitable for publication in Nature Communication or others, not in Nature Physics.

We thank Reviewer#2 for stating that they “value the new claim on the multiband CDW very highly” and for their positive opinion about the originality and novelty of our study. In the following, we respond in details to their criticism about our theoretical modeling, and explain why its phenomenological nature is a particular strength of the current analysis.

Comment 1

This system is quite simple and electron-phonon interaction or other physical quantities related with CDWs can be computed in the level of atomistic calculations. If the CDW orders are indeed multiband origins, the author can compute and can show the gap openings at some specific electronic energy bands as well as phononic dispersion from first-principles. Since the energy dependent self-consistent formulation for electronphonon coupling similar to Eliashberg formalism for the multiband superconductivity can compute in principle, the conclusive theoretical evidence for multiband CDW can be obtain with a similar approach.

We appreciate Reviewer#2’s suggestion that advanced first-principles calculations can in principle give more quantitative support for the opening of multiple gaps in NbSe₂ than our current qualitative analysis based on a simple mean-field analysis of a tight-binding band structure. However, as also explicitly acknowledged by Reviewer#1, the computation of multiband CDW formation in a quantitatively realistic model is well beyond the scope of the present study.

Moreover, in the present study, the observed energy dependence of the CDW phases directly points towards the presence of multiband CDW order.

The theoretical analysis is not meant to give a quantitative model for this effect. Rather, it is meant to show the qualitative mechanism by which band folding associated with the CDW reconstruction naturally leads to formation of (one or more) secondary gaps. Reviewer#3 makes this point clearly. The fact that this effect can be seen in a purposely simple theoretical model underlines its generic applicability. The simplicity of the model is thus a particular strength of the current analysis, as it directly suggests the same mechanism will be at play in many other CDW materials with multiple bands crossing the Fermi level. It also shows that the qualitative conclusions are independent of “small corrections”, or any particular choice of microscopic theory. Instead, the use of a simple and broadly applicable analysis establishes the generality of our results, and shows its appropriateness for the broad appeal of Nature Physics.

To address this suggestion by Reviewer#2 we include a more detailed description of our theoretical model in the methods section of the revised manuscript. We emphasize that the previous appearances of our relatively simple tight-binding model in the literature show that it already provides an excellent quantitative match with the broad range of existing experimental observations, without the need for any additional first-principles calculations. Here, we also emphasize our deliberate choice for a simple, yet broadly applicable method, with correspondingly generic appeal.

We again thank the referee for providing us with the opportunity for us to expand on the description of our theoretical model. In the longer methods section we now make it clear that our model for the CDW in NbSe₂ is constructed using a Slater-Koster fit, which takes as input local density approximation (first-principles) calculations, as well as ARPES measurements. Our model therefore agrees by construction with existing first-principles calculations.

Comment 2

Actually, I found no evidence or theoretical reasons why another CDW order occurs at -0.25 eV since the authors provide only phenomenological fitting data. So, later, it would be very persuasive and decisive work if the authors can show these data. For the moment, the concept of multiband CDWs is new and interesting but solid support for the claim is lacking.

The secondary CDW described in the current paper is not “another CDW” appearing independently of the first one. Rather, its emergence is a direct consequence of the presence of the primary CDW. As our mean-field model shows, band folding associated with the primary CDW reconstruction nat-

urally leads to the formation of (one or more) secondary gaps. This is in fact a generic, albeit often overlooked, consequence of CDW formation. This effect is clearly visible already on the phenomenological level, in 1D models, and in our mean-field analysis, and is therefore unavoidable in more detailed and more quantitative calculations. This highlights the strength of a broadly applicable phenomenological approach, giving the current analysis its broad appeal.

Reviewer#3 (Remarks to the Author):

Overview

The manuscript by Pasztor et al. reports on real-space evidence for the multiband nature of 2D charge density wave (CDW) order in an archetypal dichalcogenide material 2H-NbSe₂. By systematic model analysis of atomic-scale topography data taken with cryogenic scanning tunneling microscopy (STM) at various tip-sample bias voltages, particularly how the CDW modulations in the form of topographic STM image contrast depend on bias voltage, this work presents direct and detailed agreement of the STM topography data with a well-grounded CDW model based on three CDWs that are commensurate with the atomic lattice and on two de-phased charge modulations (CM) arising from two principal energy bands. The analytical methodology used is original, in that it extends well-established CDW formalism to multiple energy bands. The key results reported are of both breakthrough and general significance, in demonstrating a novel technique to extract real-space phase information about spatially-modulated electronic orders. Furthermore, the revelation of CDW energy gaps well away from the Fermi level represents crucial evidence for strong electron correlations in the formation of CDW order in 2H-NbSe₂.

We thank Reviewer#3 for their very positive appreciation of our study, with results of “both breakthrough and general significance” and “demonstrating a novel technique to extract real-space phase information about spatially-modulated electronic orders”.

Comment 1

In my opinion, this manuscript warrants appearance in Nature Physics, because its results shed new light on a widely-studied though still-enigmatic CDW material, and also because the novel methodology it demonstrates can

be generally applied to a variety of complex 2D materials as well as nano-engineered heterostructures (epitaxial, van der Waals or Moire pattern).

We thank Reviewer#3 for highlighting the broad applicability of our methodology and for deeming our manuscript suitable for publication in Nature Physics.

Comment 2

This work is convincing, in terms of its phenomenological clarity that connects the STM topography data to the multiband CDW model. Of particular merit is the transparent choice of model parameters, which are either constrained by the specific lattice structure of 2H-NbSe₂ (via the tight-binding band structure) or based on general physical considerations (especially that the phase shift between charge modulations of the two bands is determined by the minimization of Coulomb-repulsion as well as lattice-commensuration energies).

We thank Reviewer#3 for emphasizing the “phenomenological clarity that connects the STM topography data to the multiband CDW model” and the “particular merit” of our “transparent choice of model parameters”.

Comment 3

Experimentally, was any temperature dependence observed in the STM image contrast dependence on bias, both below and even above the CDW ordering temperature of 2H-NbSe₂? Theoretically, would one expect temperature to affect the relative strengths between the Coulomb-repulsion and latticecommensuration energies?

The temperature dependence of the CDW contrast dependence on bias would be a natural follow up study and may certainly be expected to lead to interesting and worthwhile results in itself. However, this is beyond the scope of the current manuscript, and would be very challenging (due to a combination of limited liquid helium hold time, long data acquisition time, and keeping the tip over the exact same location at all temperatures).

Arguello et al (Physical Review B 89, 235115 (2014)) have shown that the CDW is nucleating in the vicinity of defects above the CDW ordering temperature (T_{CDW}). Since we find that defects modify the bias dependence of the CDW contrast in their vicinity—with a tendency to pin the phase—, it might be difficult if not impossible to gain insight into the intrinsic behaviour of the CDW amplitude and phase in that temperature range.

We agree with Reviewer#3 that theoretically, one may expect some temperature dependence of the relative strengths of Coulomb repulsion and lattice commensuration energies. However, this is only expected to lead to small quantitative changes of the observed multigap behaviour, and does not affect the overall qualitative effect highlighted in the current manuscript.

Comment 4

What justifies neglecting the “pancake” Fermi surface (FS) sheet (centered near the Gamma point) in the multiband CDW model analysis? Was it simply because of its very small phase space relative to the other FS sheets? Or perhaps the 3D geometry of the pancake FS sheet, in contrast to the 2D tubular FS sheets, renders it less conducive to the in-plane CDW order?

We thank the reviewer for this question. Previous experimental studies see no evidence of the Se pocket being involved in the CDW. This fits with our own theoretical estimate that the bands involved in the CDW are predominantly derived from the niobium $d_{3z^2-r^2}$ orbitals while the “pancake” band is primarily Se-derived. We believe both suggestions of Reviewer#3 are legitimate further relevant considerations.

Comment 5

What does the appearance of energy gaps away from the Fermi level imply about the role that strong electron correlations play in the formation of CDWs in 2H-NbSe₂, since band structure calculations by Johannes, Mazin and Howells (Ref. 11) had shown that conventional mechanism based predominantly on Fermi surface nesting does not adequately explain the observed CDW order.

The explanation that we propose is in principle independent on the actual driving mechanism of the CDW. It only relies on the folding of the bands due to the new periodicity of a primary transition. Electron correlations enter via competition with the lattice commensuration to set the relative real space phase difference of the CMs. Our result therefore does not give any further insight in the strength of electron correlations, nor in their role in the formation of the CDW.

For completeness, we would like to point out that the issue of the band structure calculations by Johannes, Mazin and Howells, not showing a conventional peak in the Lindhard function, has already been addressed in the context of the tight-binding model used in the present study, by two of the

present authors. Including a structured electron-phonon coupling, with explicit orbital and momentum dependence, allows this model to quantitatively reproduce a broad array of experimental observations. It also naturally leads to the formation of a peak in the full electronic susceptibility at the observed CDW wave vector, which provides a mechanism for CDW formation in the absence of sharp peaks in the bare Lindhard function. A brief discussion of these previous results based on the model used here, is included in the Methods section of the revised manuscript.

Comment 6

For clarity in the caption of Figure 4, specify the color coding of the middle column of panels (b, e, h) in order to disambiguate it from that of the right column of panels (c, f, i).

We thank Reviewer#3 for drawing our attention to the potentially confusing caption of Figure 4. In our revised manuscript we replace in the caption of Fig. 4

Second column: corresponding simulated topographic traces at selected biases.

with

Second column: corresponding simulated topographic traces at selected biases (the actual bias value is shown in the legend). The red and blue topographic traces correspond to positive and negative sample biases respectively.

Comment 7

In Line 243, perhaps replace the word "backing" by "supporting".

We thank Reviewer#3 for this suggestion that we have included in our revised manuscript replacing:

These features are indicative of CDW gaps opening away from E_F in addition to the (primary) gap at E_F , backing the simple model we propose to understand the bias dependence of the CDW appearance in STM images of NbSe₂.

with

These features are indicative of CDW gaps opening away from E_F in addition to the (primary) gap at E_F , supporting the simple model we propose to understand the bias dependence of the CDW appearance in STM images of NbSe₂.

Reviewer comments, second version:

Reviewer #1 (Remarks to the Author: Overall significance):

I'd like to thank the authors for their detailed reply and some nice comparisons to literature. I have a few follow-up comments, with regards to this response:

1. I disagree with their statements about defects. Defects can induce strain, and as they aptly point out, they can affect the CDW. There are multiple types of defects in these crystals, and the PNAS reference was to show an extreme example of how it can completely modify the order. In fact, the STM cannot probe all subsurface defects, but they can contribute to the CDW. The point is simple: the claim that an experiment proves a multiband CDW requires one to study a clean material riddled with these effects. Otherwise, one has to prove that any of these additional contributions does not lead to extra modulations of the CDW, or perturbs its state. As the authors nicely show, they do not necessarily have a cleaner material than previous studies, and therefore, it is not convincing that these experiments can rule out that the defects play a role in their analysis. I can make a constructive comment: they can measure this effect in many areas and perform a statistical analysis. If only certain defects affect the CDW, then this will lead to error bars/noise in their analysis, whereas the main trends may prevail. But, I believe the role of defects on the CDW, as well as the electron-phonon coupling, and electronic structure, is vital to understand before one can make this claim. I believe it is evident, that CDW formation is sensitive to small changes of the Fermi surface, something a critical amount of defects can do.

I believe the defect analysis that has been performed is interesting (in the supplement). Likewise, I am unsure that the analysis is "far" enough away from a defect. To prove this, a disorder potential at minimum needs to be extracted, and probe that the regions studied are considered "pristine." If defects can be neglected, this needs a much stronger discussion in the paper, in any revised publication.

2. I appreciate that they describe their theoretical approach in more detail. But I still find it unsatisfactory, as did another referee. Also, to be fair, there are first principles methods that do strive to calculate CDWs (e.g. single band approaches using dual bosons, etc). I do not agree that there are no methods, although I do not think it is fair they have to do this. I believe the authors should double check. Here is one such example from three years ago: <https://doi.org/10.1038/s41535-018-0105-4>

I believe that this paper should be published after addressing the minor revisions, in Communication Physics.

Reviewer #1 (Remarks to the Author: Strength of the claims):

I do not feel the strength of the claims has improved from my first comment. However, I also think that with minor revisions, there is enough evidence to publish this. The authors should acknowledge certain points in the discussion, in the manuscript.

Reviewer #2 (Remarks to the Author: Overall significance):

Renner and coworkers have answered all questions from Referees thoroughly. I also acknowledge authors' argument on merits of their phenomenological modeling of multiband charge density waves (CDWs).

Though I found the current manuscript is better shaped than before, I still hesitate to say that a phenomenological theory of multiple-folded-bands-induced CDWs is a strong and general enough breakthrough for physics community. And, some case, I found that the application of the model and method may not be so general as the authors anticipate.

Considering, for example, that if we apply this to 2D rectangular lattice with two-bands and follow the arguments of the authors by neglecting one phase, the remaining dephasing phase should be constant.

So, all in all, I recommend a publication of the manuscript in Nature Communication.

Reviewer #3 (Remarks to the Author: Overall significance):

I have read the point-by-point rebuttals by Pasztor et al. to all three referee reports, as well as the revised manuscript which systematically addressed the key issues raised.

On the theoretical front, crucial improvement has been made in the Methods section, in fitting a multi-orbital Slater-Koster tight binding model to the band structure of 2H-NbSe₂. In my opinion, this modelling refinement soundly addresses the concerns of the other two Referees about the authors' multiband CDW interpretation being too phenomenological. The refinement also answers my own question about the justification for neglecting the Se-orbitals, manifested as the pancake sheet of the Fermi surface, in the multiband-CDW model. In fact, this negligibility distinguishes the multiband CDW from the multiband superconductivity (SC), which is known, from both STM spectroscopy and ARPES experiments, to involve the pancake sheet. This key difference, between the CDW and SC in 2H-NbSe₂, obviates the additional theoretical concern raised by Referee 2.

On the experimental side, the authors added discussion about the insensitivity of their data interpretation to the presence of atomic-scale defects, backed up by an overview comparison (Response Figure 1) of similar STM data in the literature. This discussion mitigates Referee 1's concern about the quality of the crystals measured. Having measured numerous 2H-NbSe₂ crystals in my own STM studies, supplied by several high-quality crystal growers in the field, I can objectively agree with the authors' assertion that the crystals they measured are as low in defects as it gets. And as the authors correctly point out, although local defects can in principle affect the long-range stability of CDW, it is more likely that defect-induced strain has a stronger effect. This is no doubt an important issue, and the publication of this study will only generate further interest to elucidate it.

In view of the clarifications and improvements given by Pasztor et al., I am even more convinced that this manuscript warrants publication in Nature Physics, by virtue of its phenomenological clarity for 2H-NbSe₂ and the general applicability of its methodology to other complex 2D materials as well as heterostructures.

Reviewer #3 (Remarks to the Author: Impact):

Included in Overall Significance

Reviewer #3 (Remarks to the Author: Strength of the claims):

Included in Overall Significance

Author rebuttal, second version:

“Multiband charge density wave exposed in a transition metal dichalcogenide” – Second Responses to Reviewers

Reviewer#1 :

I'd like to thank the authors for their detailed reply and some nice comparisons to literature. I have a few follow-up comments, with regards to this response:

We thank Reviewer#1 for acknowledging our detailed reply and comparison with the literature regarding defects. We stress that this comparison clearly demonstrates that our samples are as clean as NbSe₂ can be. This point is objectively confirmed by Reviewer#3 who explicitly writes that our response “...mitigates Referee 1’s concern about the quality of the crystals measured” based on that Reviewer’s own STM studies of different crystals “supplied by several high-quality crystal growers in the field”.

Comment 1

I disagree with their statements about defects. Defects can induce strain, and as they aptly point out, they can affect the CDW. There are multiple types of defects in these crystals, and the PNAS reference was to show an extreme example of how it can completely modify the order. In fact, the STM cannot probe all subsurface defects, but they can contribute to the CDW. The point is simple: the claim that an experiment proves a multiband CDW requires one to study a clean material ridden of these effects. Otherwise, one has to prove that any of these additional contributions does not lead to extra modulations of the CDW, or perturbs its state. As the authors nicely show, they do not necessarily have a cleaner material than previous studies, and therefore, it is not convincing that these experiments can rule out that the defects play a role in their analysis. I can make a constructive comment:

they can measure this effect in many areas and perform a statistical analysis. If only certain defects affect the CDW, then this will lead to error bars/noise in their analysis, whereas the main trends may prevail. But, I believe the role of defects on the CDW, as well as the electron-phonon coupling, and electronic structure, is vital to understand before one can make this claim. I believe it is evident, that CDW formation is sensitive to small changes of the Fermi surface, something a critical amount of defects can do.

I believe the defect analysis that has been performed is interesting (in the supplement). Likewise, I am unsure that the analysis is "far" enough away from a defect. To prove this, a disorder potential at minimum needs to be extracted, and probe that the regions studied are considered "pristine." If defects can be neglected, this needs a much stronger discussion in the paper, in any revised publication.

Reviewer#1 is not consistent when addressing the possible impact of defects on our investigation. In their first report, they question the quality of our samples, with a supposedly much larger defect density than previously published studies. Now, following the demonstration in our first response that the defect density in our crystals is equivalent to that in other studies (today we have to add the paper by Liu et al., Science 372, 1447–1452 (2021) to this list, who claim to observe a pair density wave, a significantly more subtle density modulation than charge order), Reviewer#1 requires us "to study a clean material ridden of these effects". This request is unreasonable. There is no defect-free material to the best of our knowledge; even Si shows numerous defects when observed by STM.

Well aware of their potential impact, we do not ignore the defects as we already pointed out in our responses to the first round of reviews. On the contrary: we address them on page 2 of the main text and in the Supplementary Information. We document and discuss the CDW contrast in the vicinity of several distinct defects in different regions, actually providing some statistical insight as newly demanded here by Reviewer#1. This analysis shows a large spread in bias dependences of the CDW contrast near defects (Suppl. Fig. 5), a spread which does actually not enable the multiband analysis we propose in these regions. On the other hand, the bias dependence of the CDW contrast is consistently the same in the defect-free regions, independent of their location in the defect landscape (Suppl. Fig. 4b). This is a a very strong indication that the defect-free regions reveal intrinsic properties of the CDW modulation in NbSe₂.

Reviewer#1 doubts that we study the CDW "far enough away from a defect", requesting us to extract a disorder potential to confirm the "regions

studied are considered pristine". Our understanding is that the analysis of the defects mentioned above fulfills this request, providing equivalent information in a different form. Our data further shows that the regions away from defects, where we extract the multiband signature of the CDW, do not show any local spectroscopic or topographic features at some specific bias that could be related to some hidden or subsurface defects. Moreover, the CDW Fourier peaks are sharp and well defined, suggesting no defects are affecting the CDW in the defect-free regions (see e.g. Chatterjee et al. Nat. Commun. 6:6313 (2015) and Jolie et al. Phys. Rev. B 99, 115417 (2019)). Finally, if strain due to the proximity to the visible defects or due to some hidden subsurface defects was affecting the clean regions we consider for our analysis, we would expect some anisotropy in the bias dependence of the CDW components (e.g. the stripes observed in PNAS 2013 110 (5) 1623-1627). This is not consistent with the perfectly homogeneous bias dependence of the amplitudes of the three CDW components illustrated in Suppl Fig. 6. All these experimental facts point to the same conclusion that our multiband analysis is neither affected by proximity to the visible defects shown in Suppl. Fig. 5 nor by invisible subsurface defects.

We have added a paragraph in the Supplementary Information Section V incorporating the above points.

Comment 2

I appreciate that they describe their theoretical approach in more detail. But I still find it unsatisfactory, as did another referee. Also, to be fair, there are first principles methods that do strive to calculate CDWs (e.g. single band approaches using dual bosons, etc). I do not agree that there are no methods, although I do not think it is fair they have to do this. I believe the authors should double check. Here is one such example from three years ago: <https://doi.org/10.1038/s41535-018-0105-4>

I believe that this paper should be published after addressing the minor revisions, in Communication Physics.

Concerning the modelling, Reviewer#1 does not really have any remaining criticism, as by their own admission the first principles calculations of the CDW order that they have in mind would be "unfair to demand" in the present context. Note that Reviewer#1 provides a citation that they believe shows the importance of these first-principles calculations to NbSe₂ (doi.org/10.1038/s41535-018-0105-4). A cursory glance at the title reveals that paper to be about NbS₂, not NbSe₂, and a thorough reading reveals that

this reference explicitly rules out a comparison to NbSe₂ before prominently citing our own theoretical model for NbSe₂ in the Methods section.

Reviewer#2 (Remarks to the Author):

Overview

Renner and coworkers have answered all questions from Referees thoroughly. I also acknowledge authors' argument on merits of their phenomenological modeling of multiband charge density waves (CDWs).

Though I found the current manuscript is better shaped than before, I still hesitate to say that a phenomenological theory of multiple-folded-bands-induced CDWs is a strong and general enough breakthrough for physics community. And, some case, I found that the application of the model and method may not be so general as the authors anticipate.

Considering, for example, that if we apply this to 2D rectangular lattice with two-bands and follow the arguments of the authors by neglecting one phase, the remaining dephasing phase should be constant.

So, all in all, I recommend a publication of the manuscript in Nature Communication.

We thank Reviewer#2 for pointing out that we have "answered all questions from Referees thoroughly" and that our revised "manuscript is better shaped than before" and suitable for publication.

The Reviewer's concern about the generality of our model is not relevant. Our study does uncover a multiband nature of the CDW in NbSe₂, which happens to develop on a triangular lattice. Analysing a two-component CDW lattice on a 2D rectangular lattice would certainly require a different approach for the reason rightfully pointed out by the Reviewer. However, it would still be possible to extract the CDW phase (with respect to the underlying atomic lattice) as a function of imaging bias and reveal a possible multiband nature of the CDW. Hence, the method may be specific to a triangular lattice, but the multiband nature of the CDW we expose experimentally for the first time by STM is of general and fundamental interest.

Reviewer#3 (Remarks to the Author):

Overview

I have read the point-by-point rebuttals by Pasztor et al. to all three referee reports, as well as the revised manuscript which systematically addressed the key issues raised.

On the theoretical front, crucial improvement has been made in the Methods section, in fitting a multi-orbital Slater-Koster tight binding model to the band structure of 2H-NbSe₂. In my opinion, this modelling refinement soundly addresses the concerns of the other two Referees about the authors' multiband CDW interpretation being too phenomenological. The refinement also answers my own question about the justification for neglecting the Se-orbitals, manifested as the pancake sheet of the Fermi surface, in the multiband-CDW model. In fact, this negligibility distinguishes the multiband CDW from the multiband superconductivity (SC), which is known, from both STM spectroscopy and ARPES experiments, to involve the pancake sheet. This key difference, between the CDW and SC in 2H-NbSe₂, obviates the additional theoretical concern raised by Referee 2.

On the experimental side, the authors added discussion about the insensitivity of their data interpretation to the presence of atomic-scale defects, backed up by an overview comparison (Response Figure 1) of similar STM data in the literature. This discussion mitigates Referee 1's concern about the quality of the crystals measured. Having measured numerous 2H-NbSe₂ crystals in my own STM studies, supplied by several high-quality crystal growers in the field, I can objectively agree with the authors' assertion that the crystals they measured are as low in defects as it gets. And as the authors correctly point out, although local defects can in principle affect the long-range stability of CDW, it is more likely that defect-induced strain has a stronger effect. This is no doubt an important issue, and the publication of this study will only generate further interest to elucidate it.

In view of the clarifications and improvements given by Pasztor et al., I am even more convinced that this manuscript warrants publication in Nature Physics, by virtue of its phenomenological clarity for 2H-NbSe₂ and the general applicability of its methodology to other complex 2D materials as well as heterostructures.

We thank Reviewer#3 for their very careful reading of our response file and revised manuscript. We are delighted with the assessment of Reviewer#3 regarding the quality of our samples, the impact of defects on our observations and with their even more convinced recommendation to publish our work in Nature Physics.